# In Vitro Induction of Interspecific Hybrid and Polyploidy Derived from *Oryza officinalis* Wall

**DOI:** 10.3390/plants12163001

**Published:** 2023-08-20

**Authors:** Meimei Tan, Ruoxin Chen, Xingran Chen, Muhammad Qasim Shahid, Xiangdong Liu, Jinwen Wu

**Affiliations:** 1State Key Laboratory for Conservation and Utilization of Subtropical Agro-Bioresources, South China Agricultural University, Guangzhou 510642, China; tanmeimei1115@163.com (M.T.); erdongre@163.com (R.C.); scau_cxr@163.com (X.C.); qasim@scau.edu.cn (M.Q.S.); xdliu@scau.edu.cn (X.L.); 2Base Bank of Lingnan Rice Germplasm Resources, South China Agricultural University, Guangzhou 510642, China; 3Guangdong Provincial Key Laboratory of Plant Molecular Breeding, South China Agricultural University, Guangzhou 510642, China; 4College of Agriculture, South China Agricultural University, Guangzhou 510642, China

**Keywords:** rice, *Oryza officinalis* Wall, tissue culture, polyploidy

## Abstract

*Oryza officinalis* Wall is a potential genetic resource for rice breeding; however, its distant genome limits its crossing ability with cultivated rice. The interspecific hybridization of *O. officinalis* and cultivated rice, establishment of its tissue culture, and induction of polyploidy are ways to improve *O. officinalis*’s poor crossability. We developed an interspecific hybrid and studied its reproductive pollen development process in this work, and the results showed that abortive pollens (81.94%) and embryo sac abnormalities (91.04%) were the key causes of its high sterility. In order to induce callus formation in interspecific hybrid explants, two different culture media, namely Chu’s N-6 medium (N6) and 1/2 Murashig and Skoog medium (1/2 MS), were employed. Additionally, two plant growth regulators (PGRs), namely 2,4-dichlorophenoxyacetic acid (2,4-D) and 6-benzylaminopurine (BA), along with L-proline (Pro) and acid hydrolyzed casein, were utilized in the experiment. The optimal N6 medium, supplemented with 2.0 mg·L^−1^ 2,4-D, produced the highest induction rate (80.56 ± 5.44)%. For callus differentiation and proliferation, the MS medium supplemented with 2.0 mg·L^−1^ BA + 0.2 mg·L^−1^ NAA produced the highest differentiation rate (75.00 ± 4.97)% and seedling emergence ratio (28.97 ± 4.67)%. The optimal combination for seedling rooting was the 1/2 MS medium supplemented with 2.0 mg L^−1^ NAA and 0.2 mg L^−1^ BA, which produced an average of 13.95 roots per plant. For polyploidy induction in the interspecific hybrid, the concentration of colchicine treatment at 400 mg·L^−1^ for three days is an ideal protocol. We devised tissue culture and interspecific hybrid polyploidy induction to overcome *O. officinalis*’ poor crossability and introduce its favorable features into cultivated rice.

## 1. Introduction

*Oryza officinalis* Wall is a diploid wild rice species belonging to the CC genome, with 2n = 24 chromosomes [1]. Regarding the three existing wild rice species in China, *O. officinalis* is endemic to the Guang Dong, Hai Nan, Guang Xi, and Yun Nan provinces. *O. officinalis* has excellent disease resistance, insect resistance, and stress tolerance compared with cultivated rice [2,3,4]. *O. officinalis* plants have strong resistance to brown planthopper, white–back planthopper, and bacterial blight resistance [5,6]. *O. officinalis* also has the potential for fast growth and strong resistance; it has a potential role in germplasm improvement and novel variety breeding. Interspecific hybridization and backcross-selection represent prominent methodologies employed to effectively amalgamate the dominant genes of *O. officinalis* and cultivated rice, hence facilitating the successful breeding of novel germplasm [7,8]. To utilize the potential excellent genes of wild rice resources, conventional hybridization, embryo rescue, embryo culture, protoplast fusion, and heterologous additional lines are frequently used in wild rice [9,10]. The hybrids formed between *O. officinalis* and cultivated rice exhibit reproductive isolation, leading to significant male and female sterility. This reproductive barrier restricts the compatibility of remote hybridization and limits gene exchange and germplasm exploitation due to hybrid sterility [11]. Homologous chromosome pairing, abnormal megaspores, and abnormal microspores in the meiosis process could result in the limited utilization of *O. officinalis* [11]. Twenty-five monosomic alien addition lines (MAAL) were generated through the interspecific hybridization of *O. officinalis* and the successive backcrossing of hybrid progenies [12].

Plant tissue culture is the utilization of its cell totipotency. Under suitable conditions, plant organs, tissue cells, and protoplast can develop into a complete plant through differentiation and regeneration. Rice genotypes, explants type, basic media, PGRs, culture duration, temperature, photoperiod, and other factors can affect the rice callus formation and plant regeneration ability [13,14]. Among the different types of explants, the induction rate of rice seeds, young embryos, and young panicles is higher than that of anthers [15]. In addition, plant genotypes have a great influence on rice tissue culture. The cultivation of *japonica* rice is comparatively more feasible than that of *indica* rice, whereas cultivated rice, in general, presents greater ease of cultivation when compared to wild rice [16,17]. Leaf sheaths and buds were used to induce regenerated plants through the callus pathway and tissue culture seedlings in *O. officinalis* [18].

Polyploidy induction is an efficient strategy for improving agronomic traits and resistance in cultivated and wild plants [19,20,21,22,23]. Polyploidy induction methods could be divided into physical, chemical, and biological induction. The polyploidy induction of many crops, flowers, and fruits, such as *Catharanthus roseus* L., *Pyrus communis* L., and *Citrus sinensis*, has been frequently reported using colchicine [8,24,25]. Doubled haploid (DH) technology is also a very convenient approach for polyploidy induction in plant breeding, which can save time and corresponding costs [26,27]. A tetraploid fennel was induced by diploid fennel seeds with 0.05% (*w*/*v*) colchicine for 24 h [28]. Polyploidy was generated using lavender tissue culture seedlings into a subculture supplemented with 2% DMSO and 0.2–0.4% colchicine from 48 to 72 h.

*O. officinalis* accumulated the number of dominant genes that were absent or lost in cultivated rice during the long-term natural adaptation and selection process [2,3]. Interspecific hybridization is one of the main approaches for developing rice varieties by incorporating the dominant genes of *O. officinalis* into cultivated rice. In order to harness advantageous genetic traits found in *O. officinalis*, many techniques, like traditional hybridization, embryo rescue, embryo culture, and protoplast fusion, are commonly employed [9,10]. However, very little is known about the reproductive characteristics and tissue culture system of *O. officinalis*. In the present study, we constructed an interspecific hybrid of *O. officinalis* and cultivated rice, and observed its reproductive features using cytological analysis. Then, we developed an interspecific hybrid generated from *O. officinalis* using tissue culture and polyploidy induction. These results provide a theoretical basis for the further improvement of rice tissue culture technology, offering the possibility of transferring desirable traits into cultivated rice.

## 2. Results

### 2.1. Reproductive Characteristics of the Interspecific Hybrid Derived from O. officinalis and Cultivated Rice

This study used interspecific hybrid plants derived from *O. officinalis* and cultivated rice to investigate the reproductive characteristics of an interspecific rice hybrid constructed from *O. officinalis* (female parent, CC genome) and cultivated rice (male parent, AA genome). We detected the pollen and embryo sac fertility of interspecific hybrid plants, and found a high percentage of sterility in the interspecific hybrid (Figure 1). A total of 81.94% of abortive pollen grains was observed in interspecific hybrid plants, compared with its cultivated rice parent. Typical abortive pollens, stained abortive pollens, and spherical abortive pollens were frequently observed (Figure 1a–c). Additionally, we assessed the fertility of the embryo sac in interspecific hybrid plants. The sterility value of the embryo sac was found to be 91.04%. The main kind of embryo sac abortion seen in interspecific hybrid plants in this study was the degeneration of the megaspore mother cell (Figure 1d–f).

The cultivated rice parent displayed a normal pollen development process divided into eight stages, including the pre-meiotic stage, meiosis stage, early microspore stage, middle microspore stage, late microspore stage, early bicellular pollen stage, late bicellular pollen stage, and mature pollen stage (Appendix A). Compared with its parent, pollen development also experienced similar stage divisions. Several kinds of abnormalities in differential pollen development stages, such as abnormal pollen mother cells (PMCs), abnormal dyads, abnormal tetrads, abnormal microspores, and abnormal bicellular pollens, were observed in the interspecific hybrid plants during the pollen development process (Figure 2). Compared with its parent, several abnormal embryo sacs were also observed in the interspecific hybrid during the embryo sac development process (Figure 3). Megaspore mother cell degeneration, functional megaspore degeneration, and a small embryo sac (less than 2/3 of the normal embryo sac) also accounted for a certain proportion found in the hybrid plants (Figure 3). All of these anomalies resulted in substantial sterility rates in the hybrid plants.

### 2.2. Tissue Culture System of the Interspecific Hybrid Derived from O. officinalis and Cultivated Rice

To investigate the callus induction rate of the interspecific hybrid derived from *O. officinalis* and cultivated rice, four combinations of basic media, 2,4-D, Pro, and acid hydrolyzed casein, were used (Table 1). Differential medium combinations showed significant variations in callus induction (Figure 4). In this study, the rate of callus induction without Pro and acid-hydrolyzed casein showed the highest induction rate (80.56 ± 5.44)% (Table 1, Figure 4a–c). In comparison, the N6 medium supplemented with 2,4-D (2.0 mg·L^−1^) + acid-hydrolyzed casein (0.3 g·L^−1^) + Pro (3.0 g·L^−1^) had the lowest induction rate ratio (7.78 ± 2.35)% (Table 1, Figure 4g–i). These results indicated that the N6 medium, supplemented with 2.0 mg·L^−1^ 2,4-D, without Pro and acid-hydrolyzed casein, was more suitable for the induction of the young panicle callus.

Due to the limitation of the callus, subculture and callus proliferation were conducted using the N6 medium supplemented with 2,4-D (2.0 mg·L^−1^) and different concentrations of BA (0.0, 0.2 and 0.5 mg·L^−1^) (Figure 5, Appendix A). These results demonstrated that the N6 medium supplemented with 2,4-D (2.0 mg·L^−1^) + BA (0.2 mg·L^−1^) for subculture had a significant effect on the callus proliferation rate, and increases the number of calli (Figure 5g–i, Appendix A).

To improve the rate of callus differentiation and proliferation, the MS medium with three differential PGRs, including BA, kinetin (KIN), and α-naphthalene acetic acid (NAA), were used in this study (Table 2, Figure 6). BA combined with NAA showed better effects on callus bud differentiation. The F2 combination of the MS medium supplemented with BA (2.0 mg·L^−1^) + NAA (0.2 mg·L^−1^) demonstrated the highest differentiation and proliferation rate, with a 75% differentiation rate in the interspecific hybrid and a seedling emergence ratio of 28.97% (Figure 6d–f). These results suggest that the MS medium is more suitable for callus differentiation in the interspecific hybrid derived from *O. officinalis* and cultivated rice.

### 2.3. Influence of Exogenous Hormone Ratio on Shoot Rooting

To evaluate the relationship between PGRs and rooting formation, three 1/2 MS basic media, combined with different concentrations of BA, NAA, 3-indole acetic acid (IAA), and activated charcoal (AC), were used (Table 3, Figure 7 and Appendix A). In this study, the 1/2 MS basic medium combined with IAA, NAA, or BA promoted the rooting of seedlings. The G1 medium + NAA (2.0 mg·L^−1^) + BA (0.2 mg·L^−1^) had the optimal effect on the root seedlings, with an average of 13.95 roots per plant (Table 3, Figure 7a,c).

### 2.4. Induction of Interspecific Hybrid Derived from O. officinalis and Cultivated Rice by Colchicine

To induce polyploidy in interspecific hybrid plants, a colchicine-treated callus was utilized in this study. The callus survival rate and seedling emergence rate of colchicine co-cultured for 5 days at the same concentration were significantly lower than those for 3 days at the same concentration. We detected that 400–500 mg·L^−1^ of colchicine treatment showed a higher rate of seedling induction at 3 days compared with the 5 days. In comparison to other colchicine treatment concentrations, the 400 mg/L concentration demonstrated a better percentage of callus survival and seedling emergence (Appendix A).

Higher concentrations of colchicine treatment led to a high percentage of callus browning (Figure 8). In this study, callus browning gradually appeared after culturing for 2 to 3 days. The death of the callus was easy to detect after culturing for 20 d. Among the four concentrations of colchicine treatment, the 400 mg·L^−1^ colchicine treatment also showed a low percentage of callus browning (Appendix A). During the differentiation and re-differentiation process, we detected that the differentiation rate of the callus under colchicine treatment was slower than the no-treatment condition. Over a period of 2 to 20 days, the callus underwent a gradual process of browning and subsequent death (Figure 8, Figure 9 and Appendix A). Differentiated seedlings treated with colchicine showed thicker and abnormal-shape leaves.

### 2.5. Investigation of the Agronomic Traits of Colchicine-Induced Plants

The agronomic traits of the plants obtained from tissue culture after colchicine treatments were investigated. Compared with its original material, candidate materials of mixed-ploidy plants were obtained in this study (Figure 10). After treatment, the mixed-ploidy material showed no significant difference in average plant height, leaf length, leaf width, and stem diameter. Leaf width and plant thickness were consistent with the general morphological rule of polyploid plants (Table 4). However, its tiller number increased significantly, contrary to the tiller reduction characteristic in most polyploid rice genotypes.

## 3. Discussion

### 3.1. Low Fertility Is the Major Limitation for the Utilization of the Hybrid Derived from O. officinalis and Cultivated Rice

Distant hybridization caused the recombination of highly heterologous genomes [13]. The incompatibility of chromosomes frequently led to the low fertility of distant hybrids, which is a significant barrier for their potential use. Pollen sterility and embryo sac abortion are important factors affecting distant rice hybrid fertility. Therefore, understanding the reproductive characteristics of interspecific hybrids is an important factor for using the potential germplasm of distant rice hybrid plants. Until now, very little has been known about the reproductive characteristics of interspecific hybrid plants of *O. officinalis*. From this study, we constructed hybrid plants of *O. officinalis* and cultivated rice to utilize the excellent gene resources of *O. officinalis*. Different types of abnormalities were detected in an interspecific hybrid of *O. officinalis* during rice pollen and embryo sac development. Abnormalities of the embryo sac, such as functional megaspore degradation, female reproductive unit degradation, and dysplasia, are frequently detected in *O. officinalis* × cultivated rice hybrids [29]. Our results proved the incompatibility of distant hybridization between *O. officinalis* and cultivated rice.

### 3.2. Factors Influencing Tissue Culture in the Interspecific Hybrid Derived from O. officinalis and Cultivated Rice

The type of explants, genotypes, exogenous PGRs, and other additives are important factors that affect the rice tissue culture. Rice anthers, young panicles, young embryos, and mature embryos were frequently used as explants in tissue culture [30]. From this study, high sterility was the major hindrance to the interspecific hybrid of *O. officinalis* and cultivated rice, leading to the difficulty in obtaining seed embryos. Therefore, we used young panicles as the explants, and preliminarily established the tissue culture of the interspecific hybrid of *O. officinalis* and cultivated rice. We screened the suitable culture formula of the interspecific hybrid, and the maximum emergence rate reached 28.97%. The emergence rate of this study was close to the other *O. officinalis* material in the Yunnan province [31].

Medium components have a great effect on interspecific hybrid callus induction. The nutrient broth (NB) basic medium produced the highest induction rate for the *Oryza alta* Swallen [23]. The N6 medium is suitable for interspecific hybrid callus induction, which was the same as the basic media for the callus induction of *indica* and *japonica* rice [32,33]. However, we detected that the MS medium was suitable for the callus induction culture of *O. officinalis*. This result differs from previous results and indicates that the suitable basic medium for induction probably has some relationship with rice genotypes.

PGRs and other additives are frequently used to improve the callus induction rate in interspecific hybrids. The type of exogenous PGRs added depends on the material. For example, 2,4-D can effectively promote callus formation in rice callus induction [34]. A 2.0–3.0 mg·L^−1^ 2,4-D concentration was suitable for callus induction in *indica* rice [35]. The combination of 6.0 mg·L^−1^ 2,4-D with NAA, IAA, and KIN was used to induce wild rice callus [36]. In the *O. officinalis*, the optimal medium for callus induction was 1.0 mg·L^−1^ 2,4-D combined with a small amount of cytokinin analogs [18]. From this study, 2.0 mg·L^−1^ 2,4-D was considered the most suitable concentration for callus induction in interspecific hybrid of *O. officinalis* and cultivated rice without adding other exogenous PGRs. The concentration of 2,4-D of interspecific hybrid was similar to the induction of *indica* rice and other wild rice material. All of these results indicated that 2,4-D played an important role in the induction of rice callus.

Cytokinins, such as BA or KIN, are essential and frequently added in the differentiation process. In the tissue culture of *O. alta*, 2.0 mg·L^−1^ KIN was used for callus differentiation, and the dosage was 10 times that of auxin analogue NAA [23]. In the *O. officinalis*, 2.0–4.0 mg·L^−1^ BA + 1.0–2.0 mg·L^−1^ KIN combined with 1.0 mg·L^−1^ IAA was the most suitable for differentiation and growth [31]. In this study, 2 mg·L^−1^ NAA combined with 0.2 mg·L^−1^ BA had the optimal effect in promoting differentiation. The ratio of exogenous PGRs was similar to Zhang et al. [23]. These results indicated that medium components, PGRs, and cytokinins play a critical role in the tissue culture of interspecific hybrids.

### 3.3. Polyploidy Induction of Interspecific Hybrids Derived from O. officinalis and Cultivated Rice

Colchicine-induced tetraploids have proven effective and were successfully employed in many plants [37,38]. A proper dosage range and treatment method of colchicine, as a frequently utilized polyploidy inducer substance, can successfully generate polyploidy plants [39,40,41]. However, colchicine treatment caused callus damage, resulting in the browning and death of the callus, and reduced differentiation ability. In the Gramineae plant, the induced polyploidy concentration of colchicine is always from 200 to 800 mg·L^−1^ [42]. This study found that 400–500 mg·L^−1^ of colchicine treatment showed a higher rate of 3-day seedling induction, which was comparable to earlier research on *O. alta* [42]. We speculated that the toxicity of colchicine in rice is similar within a specific range of concentration, and the toxicity has an enhanced effect when the concentration is higher than a certain limit.

Colchicine treatment led to the irreversible death of the callus cells [43]. High colchicine concentration always produces a large amount of toxins during the induction process. This study found that colchicine concentration was related to seedling emergence in interspecific hybrid plants. The toxic effect of colchicine is significantly increased when the colchicine concentration reaches 600 mg·L^−1^. This result was similar to the polyploidy induction analysis of *O. alta*. The callus survival rate and seedling emergence rate of *O. alta* under the treatment of 600 mg·L^−1^ colchicine for 24 h was much higher than that of 200 to 400 mg·L^−1^ colchicine treatment [23]. In addition, we found that the colchicine treatment time also directly influenced the toxic effect in this study. Co-culturing for 5 days in colchicine treatment showed a stronger toxic effect than co-culturing for 3 days. This result is consistent with a previous study, which revealed that polyploidy induction of gerbera under colchicine treatment at 1% for 4 and 8 h showed a highly toxic effect compared to the treatment of 0.1% colchicine for 8 h [44]. All of these results indicated that colchicine concentration and treatment duration play a significant role in the induction of polyploidy in interspecific hybrids derived from *O. officinalis* and cultivated rice.

Polyploidy material exhibited non-significant differences in average plant height, leaf length, and stem diameter following treatment. Conforming to the general morphological norm of double-chromosome plants, leaf width and plant thickness were consistent. However, the number of tillers increased substantially compared to the tiller reduction typical of most polyploid rice genotypes [45].

## 4. Materials and Methods

### 4.1. Plant Material and Plant Growth Conditions

This investigation employed one interspecific hybrid created by *O. officinalis* and cultivated rice. The *O*. *officinalis* belongs to the CC genome and was utilized as the female parent, whereas cultivated rice belongs to the AA genome and was used as the male parent. All materials were grown at the farm (23°15′ N, 113°21′ E) of South China Agricultural University (SCAU), and management practices were according to the recommendations for the area. During the growing season, the average temperature was between 23 and 29 °C, with a relative humidity range of 74 to 88%. The young panicles of interspecific hybrid plants were collected and used for tissue culture, and the callus was employed in chromosome-doubling experiments by colchicine.

### 4.2. Cytological Observation of the Pollen and Embryo Sac Development Process

To verify pollen and embryo sac fertility in interspecific hybrid plants, whole-mount eosin B-staining confocal laser-scanning microscopy (WE-CLSM) was used. The young panicles were collected and fixed in an FAA solution for 48 h. Then, the samples were washed using 95% and 80% ethanol for 30 min, and kept in 70% ethanol at 4 °C until observation. Interspecific hybrid plant pollen and embryo sac fertility were detected with minor modifications to our previous research [46,47].

### 4.3. Tissue Culture of the Interspecific Hybrid Derived from O. officinalis

The young panicles of interspecific hybrid plants with a 0.5–5.0 cm range between their flag leaf cushion and the second to last leaf cushion were collected and stored in a refrigerator at 4 °C. The panicles were surface-sterilized with 75% ethanol for 30 s, and then disinfected in 0.1% HgCl_2_ (mercury chloride) solution for 8–10 min. Then, samples were rinsed with sterile distilled water 4–6 times. Then, using five basic induction media with N6 or 1/2MS as the basic media and varying amounts of exogenous PGRs, immature panicles with 3.0 mm segments were treated, including 2,4-D (2.0 mg·L^−1^), Pro (0, 0.3, and 3.0 g·L^−1^) and acid-hydrolyzed casein (0, and 0.3 mg·L^−1^).

To produce high-quality of calli, the N6 medium supplemented with 2,4-D (2.0 mg·L^−1^) and three concentrations of BA (0, 0.2, and 0.5 mg·L^−1^) were used for the subculture. All media were placed in the incubator at 28 °C and dark-cultured for 26 d. After the callus subculture, the basic medium of the MS and different concentrations of BA (0 and 2.0 mg·L^−1^), KIN (0 and 2.0 mg·L^−1^), and NAA (0.2, 0.4, and 1.0 mg·L^−1^) were used for differentiation. The 1/2MS medium supplemented with 0.5 g·L^−1^ activated charcoal (AC) and different concentrations of IAA (0, 1.0, and 2.0 mg·L^−1^), NAA (0, 1.0, and 2.0 mg·L^−1^), and BA (0.2, and 0.5 mg·L^−1^), was added to prepare the medium for promoting the rooting and vigorous seedlings.

### 4.4. Callus Treatment with Colchicine Solution

To induce an allopolyploid interspecific hybrid, the callus and cluster buds immersed in the liquid medium, which contained 1.5% dimethyl sulfoxide (DMSO) and five differential concentrations of colchicine (0, 300, 400, 500, and 600 mg·L^−1^). The light yellow undifferentiated or partially green calli were immersed in a colchicine solution and cultured in a dark environment under a 150 rpm shaking table at 25 °C for 24 h. The treated callus was washed with sterile water and inoculated in the MS medium, including the different concentrations of BA (0, and 2.0 mg·L^−1^), KIN (0, and 2.0 mg·L^−1^), and NAA (0.2, 0.4, and 1.0 mg·L^−1^) for differentiation. They were cultured under dark conditions at 28 °C for 2 days, and then the samples were transferred to a light-controlled environment to facilitate the development of seedlings.

### 4.5. Investigation of the Allopolyploid Hybrid and Statistical Analysis

To detect the variations in candidate polyploid plants, agronomic traits, including plant height, number of tillers, leaf shape, and stem diameter, were observed in this study. In accordance with the prior investigation [48], all characteristics were identified and subjected to statistical analysis. Excel 2016 and IBM SPSS Statistics 21.0 were used for data statistics and analysis in this study. The flow cytometric analysis was conducted based on our earlier research, with minor modifications [23].

## 5. Conclusions

The findings of this study suggest that the key factor contributing to the low fertility of the plant under investigation is the occurrence of pollen abortion and embryo sac abnormalities, as evidenced by the cytological analysis results. A tissue culture system has been generated for an interspecific hybrid derived from *O. officinalis* and cultivated rice. Moreover, the colchicine treatment at 400 mg·L^−1^ for 2 days is the optimal protocol to induce polyploidy in rice. The induction of polyploidy in *O. officinalis* and the tissue culture system described in this work may prove advantageous in addressing the issue of limited cross-compatibility in *O. officinalis*.

## Figures and Tables

**Figure 1 plants-12-03001-f001:**
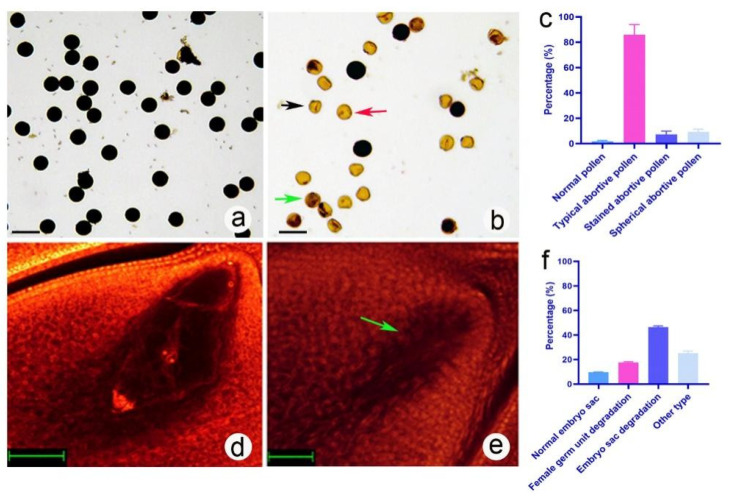
Comparison of the pollen and embryo sac fertility between the interspecific hybrid and its parent. (**a**) The pollen phenotype of cultivated rice; (**b**) the pollen phenotype of the interspecific hybrid (the black arrow indicates typical abortive pollens, the green arrow indicates stained abortive pollens, and the red arrow indicates spherical abortive pollens); (**c**) pollen fertility value of the interspecific hybrid; (**d**) embryo sac phenotype of cultivated rice; (**e**) abortive embryo sac of interspecific hybrid, the arrow indicates embryo sac degeneration; (**f**) embryo sac fertility value of the interspecific hybrid. Bar = 40 μm.

**Figure 2 plants-12-03001-f002:**
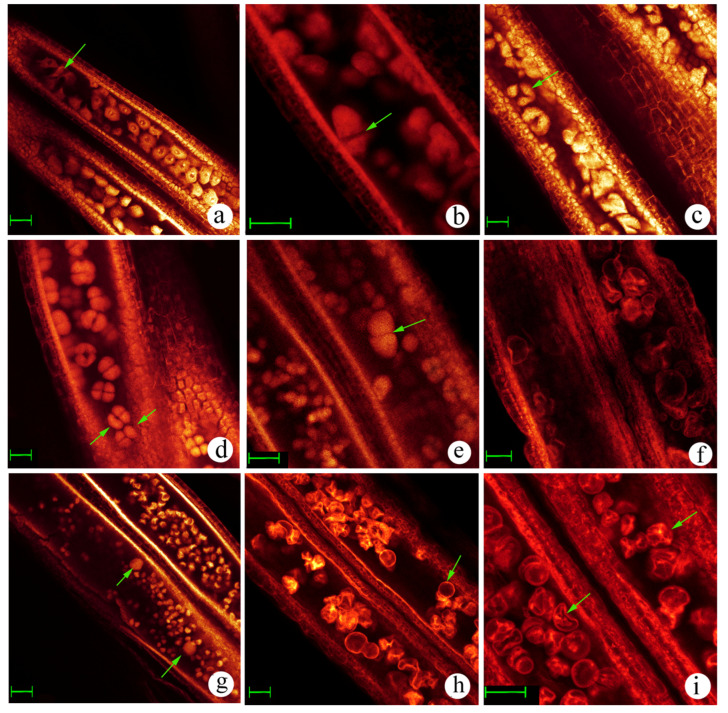
The pollen development process of interspecific hybrid. (**a**) Pre-meiotic stage (the arrow indicates the degraded pollen mother cell); (**b**) telophase I (the arrow indicates asynchronous pollen); (**c**) telophase II (the arrow indicates abnormal tetrad cell, triad); (**d**) tetrad stage (the arrows indicate abnormal tetrad cells, two triads); (**e**) tetrad stage (the arrow indicates the abnormal shape of the tetrad); (**f**) single microspore stage, degradation of microspore cells; (**g**) Middle single microspore stage (the arrows indicate the abnormal formation of microspores); (**h**) mature pollen stage (the arrow indicates spherical abortive pollen); (**i**) mature pollen stage (the arrows indicate typical abortive pollens). Bar = 40 μm.

**Figure 3 plants-12-03001-f003:**
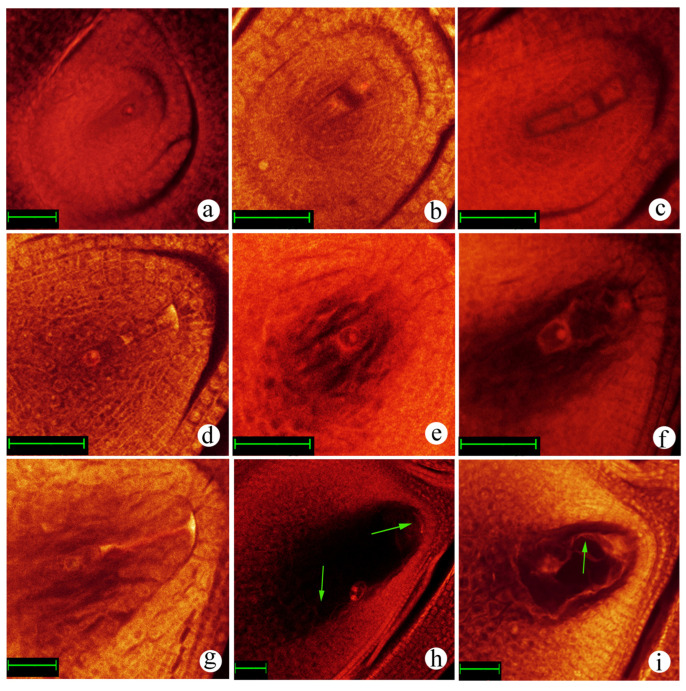
Embryo sac development of the interspecific hybrid. (**a**) Megasporocyte stage; (**b**) meiotic division stage, dyad stage; (**c**) meiotic division stage, tetrad stage; (**d**) functional megaspore stage; (**e**) mono-nucleate embryo sac stage; (**f**) two-nucleate embryo sac stage; (**g**) meiotic division stage, tetrad stage, abnormal degradation of megaspore cell; (**h**) mature embryo sac stage (the arrows indicate female germ unit and polar nuclei degradation); (**i**) mature embryo sac stage (the arrow indicates no female germ unit). Bar = 40 μm.

**Figure 4 plants-12-03001-f004:**
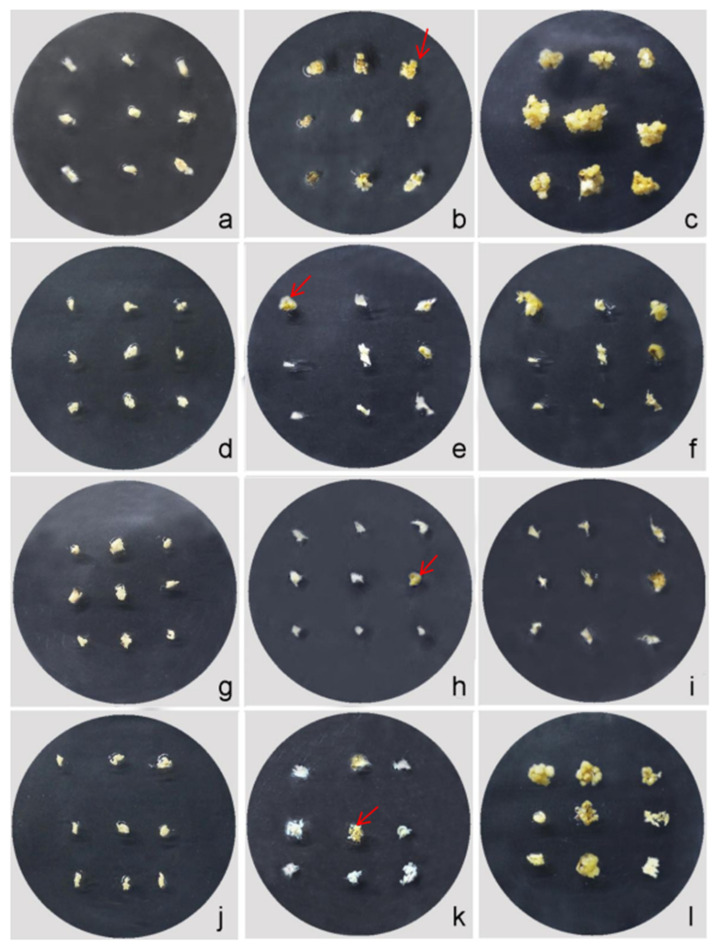
Callus induction of the interspecific hybrid in different media. (**a**–**c**) Callus. Induction in the Y1 medium at 0 d, 10 d, and 30 d, respectively. (**d**–**f**) Callus induction in the Y2 medium at 0 d, 10 d, and 30 d, respectively. (**g**–**i**) Callus induction in the Y3 medium at 0 d, 10 d and 30 d, respectively. (**j**–**l**) Callus induction in the Y4 medium at 0 d, 10 d and 30 d, respectively. The red arrow indicates the initiation of differentiation.

**Figure 5 plants-12-03001-f005:**
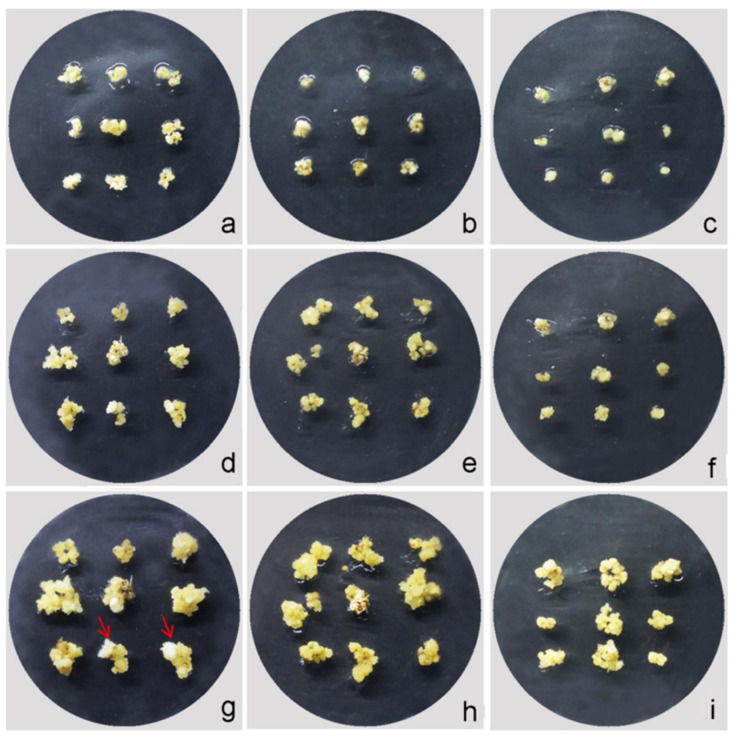
Effect of different BA concentrations on the callus proliferation of the interspecific hybrid. (**a**,**d**,**g**) Callus was sub-cultured in the J1 medium for 2 d, 10 d, and 25 d, respectively. (**b**,**e**,**h**) Callus was sub-cultured in the J2 medium for 2 d, 10 d, and 25 d, respectively. (**c**,**f**,**i**) Callus was sub-cultured in the J3 medium for 2 d, 10 d, and 25 d, respectively. The arrow indicates embryogenic degeneration of the callus with white color.

**Figure 6 plants-12-03001-f006:**
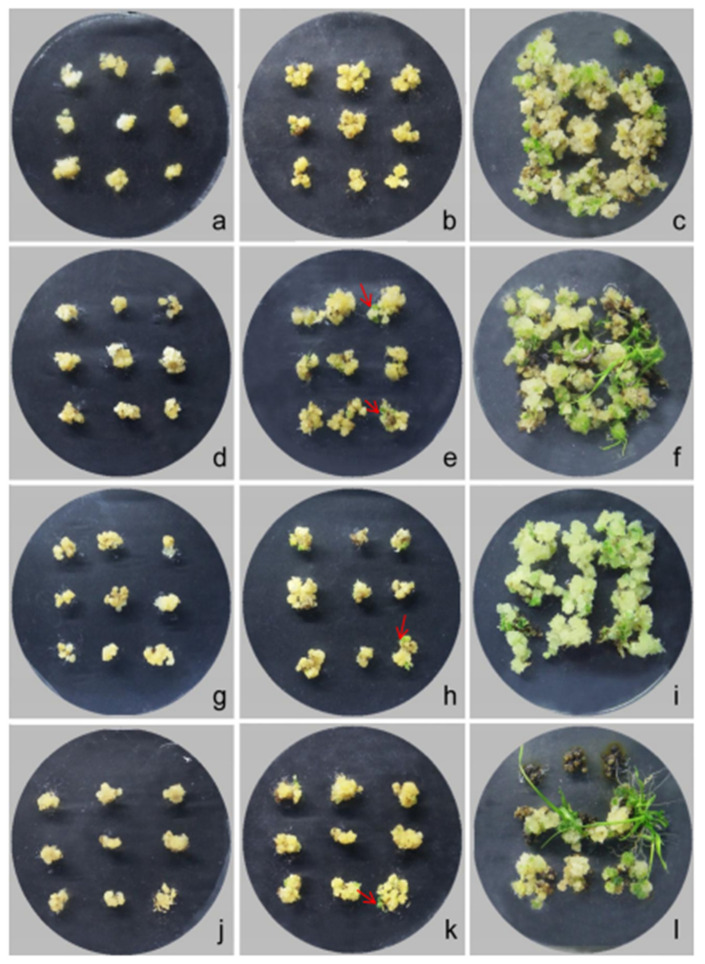
Effect of the exogenous hormone ratio on callus differentiation. (**a**–**c**) Callus was differentiated and cultured in the F1 medium for 2 d, 9 d, and 40 d, respectively. (**d**–**f**) Callus was differentiated and cultured in the F2 medium for 2 d, 9 d, and 40 d, respectively. (**g**–**i**) Callus was differentiated and cultured in the F3 medium for 2 d, 9 d, and 40 d, respectively. (**j**–**l**) Callus was differentiated and cultured in the F4 medium for 2 d, 9 d, and 40 d differentiation of callus. The arrow points to the differentiated green dot.

**Figure 7 plants-12-03001-f007:**
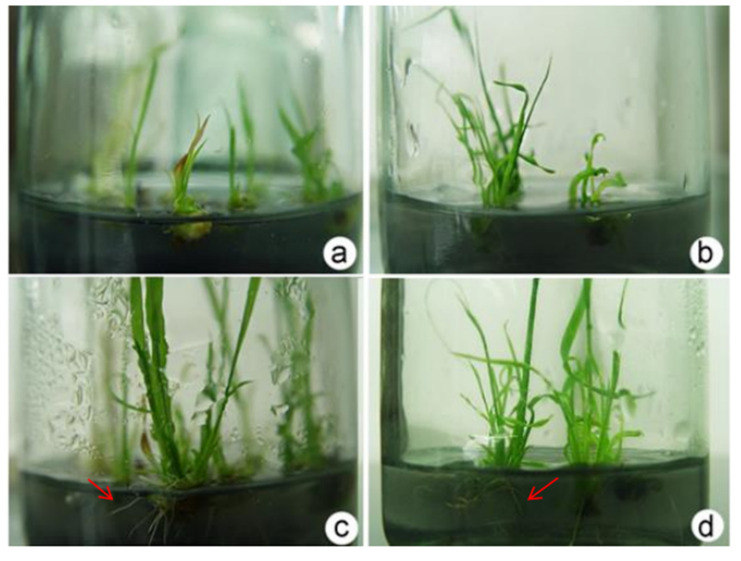
Tissue culture of strong rooting seedlings derived from the callus of the interspecific hybrid. (**a**,**c**) Root-strong seedlings cultured in the G1 medium for 1 d and 10 d, respectively. (**b**,**d**) Cultured in the G2 medium for 1 d and 10 d, respectively. The arrow indicates the rooting of the plants.

**Figure 8 plants-12-03001-f008:**
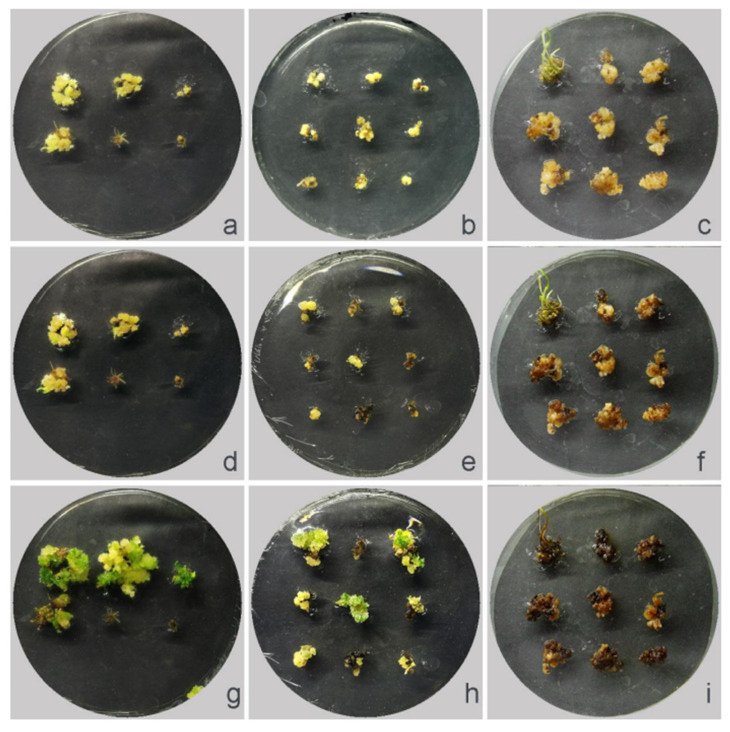
Callus status after co-culturing with different colchicine concentrations. (**a**,**d**,**g**) Callus was co-cultured with 400 mg·L^−1^ colchicine for 3 days and differentiated for 2 d, 10 d, and 20 d, respectively. (**b**,**e**,**h**) Callus was co-cultured with 500 mg·L^−1^ colchicine for 3 days and differentiated for 2 d, 10 d, and 20 d, respectively. (**c**,**f**,**i**) Callus was co-cultured with 600 mg·L^−1^ colchicine for 3 days and differentiated for 2 d, 10 d, and 20 d, respectively.

**Figure 9 plants-12-03001-f009:**
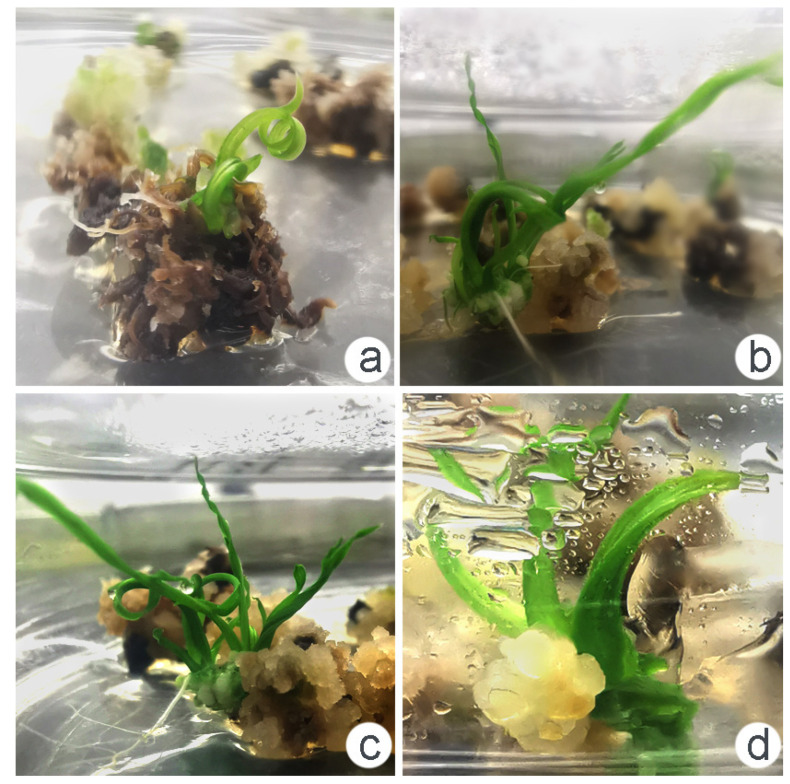
Growth status of mutant seedlings obtained through colchicine treatment. (**a**–**c**) The crouching and twisting leaves under colchicine treatment; (**d**) abnormal stout seedling under colchicine treatment.

**Figure 10 plants-12-03001-f010:**
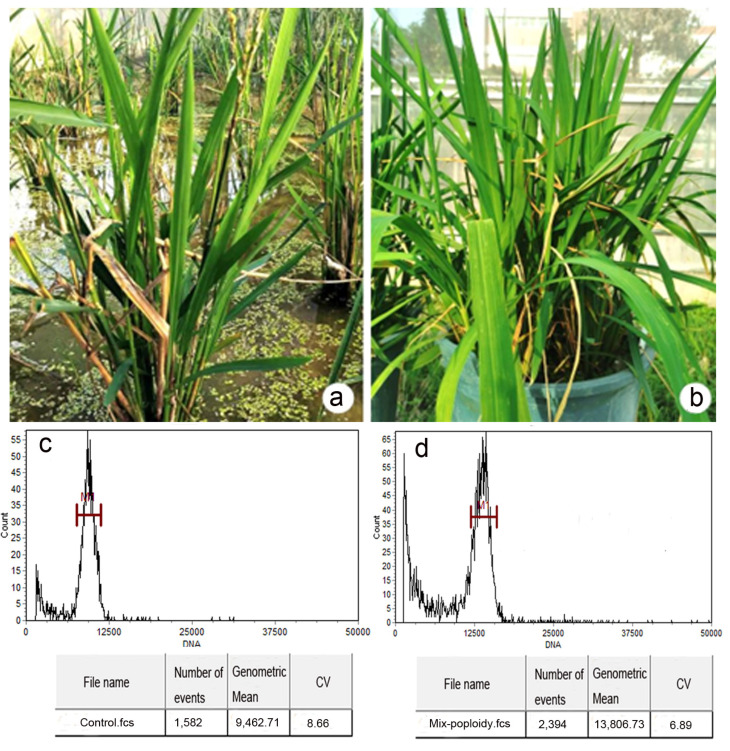
Comparison of mixed-ploidy plant and its original interspecific hybrid plant. (**a**) The phenotype of the interspecific hybrid plant; (**b**) the phenotype of the mixed-ploidy plant induced by colchicine treatment; (**c**) flow cytometric analysis of the interspecific hybrid plant; (**d**) flow cytometric analysis of induced candidate polyploid plants.

**Table 1 plants-12-03001-t001:** Callus induction rate of different media and plant growth regulators.

Combination Name	Medium Name	2,4-D Concentration (mg·L^−1^)	Acid-Hydrolyzed Casein Concentration (g·L^−1^)	Pro Concentration (g·L^−1^)	Induction Rate (%)
Y1	N6	2.0	0.0	0.0	80.56 ± 5.44 a
Y2	N6	2.0	0.3	0.3	18.52 ± 3.17 bc
Y3	N6	2.0	0.3	3.0	7.78 ± 2.35 c
Y4	1/2MS	2.0	0.0	0.0	30.14 ± 5.18 b

Note: Data are shown as the mean and standard error (SE). Different letters represent significant differences at *p* < 0.01.

**Table 2 plants-12-03001-t002:** Effects of exogenous hormone ratio on callus differentiation.

Combination Name	Differentiation Rate (%)	Seedling Emergence Rate (%)	Browning Rate (%)
F1	46.46 ± 7.61 b	12.12 ± 5.70 B	24.24 ± 4.93 B
F2	75.00 ± 4.97 a	28.97 ± 4.67 A	14.29 ± 3.38 B
F3	50.56 ± 6.25 b	9.72 ± 4.84 B	8.80 ± 3.71 B
F4	47.62 ± 8.11 b	5.56 ± 2.26 B	49.21 ± 8.54 A

Note: Data are shown as the mean and standard error (SE). Different capital letters in the same column represented significant differences at *p* < 0.01, and lowercase letters in the same column represented significant differences at *p* < 0.05.

**Table 3 plants-12-03001-t003:** Influence of exogenous hormone ratio on the number of roots of the shoots.

Combination Name	Medium Name	IAA Concentration (mg·L^−1^)	NAA Concentration (mg·L^−1^)	BA Concentration (mg·L^−1^)	AC Concentration (g·L^−1^)	Root Number
G1	1/2MS	0.0	2.0	0.2	0.5	13.95 ± 1.04 a
G2	1/2MS	2.0	0.0	0.2	0.5	11.85 ± 1.72 a
G3	1/2MS	1.0	1.0	0.5	0.5	10.77 ± 1.25 a

Note: Data are shown as the mean and standard error (SE). Different lowercase letters represent significant differences at *p* < 0.05.

**Table 4 plants-12-03001-t004:** Comparison of agronomic traits between colchicine-induced plants and original materials.

Materials	Tiller Number	Plant Height (cm)	Leaf Length (cm)	Leaf Width (cm)	Stem Diameter (cm)
Control	8.67 ± 0.92	74.63 ± 1.21	48.88 ± 0.72	1.42 ± 0.04	0.44 ± 0.02
Mixed-ploidy plant	29.29 ± 3.06 **	70.99 ± 1.15	38.8 ± 1.54 **	1.91 ± 0.12 **	0.63 ± 0.05 *

Note: Control represents the original interspecific hybrid. * and ** represent statistically significant differences at *p* < 0.05 and *p* < 0.01, respectively.

## Data Availability

Not applicable.

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
