# Peer review of "In Vitro Induction of Interspecific Hybrid and Polyploidy Derived from *Oryza officinalis* Wall"

_plants, 2023, doi:10.3390/plants12163001_

Round 1

Reviewer 1 Report

This manuscript studied rice Oryza officinalis Wall tissue culture technology, observed pollen development, obtained callus and regenerated plantlets, and obtained some polyploid seedlings by colchicine treatment at 400 mg/L, the manuscript on flow cytometry data and field cultivation seedlings traits observation is very important, although the data is not quite perfect, I hope the author will make further observations, please transplant them into the text;

The manuscript writing is rough. Careful examination and revision are required.

The rice Oryza officinalis Wall appears hundreds of times in the manuscript. It appears for the first time, the full name is required, after which it is referred to written as O. officinalis; The same is Oryza alta Swallen.

Spaces are needed in many places in the manuscript, such as between numbers and units, time and weight units, etc. Spaces are required before citations Instead of PGRs;

The tissue culture process is not hormone, but plant growth regulators (PGRs); When plant growth regulators first appear, they need to be called in full, including in the abstracts and text, and then they can be abbreviated;

6-BA changed as BA, KT changed as KIN; mg/L changed as mg L-1, activated carbon changed as activated charcoal;

To avoid using best in the manuscript, there is no best, only the change sign, it is recommended to change to “optimal”;

The reviewers marked the PDF version with a yellow marker, wish the authors can revise them.

Need English improving

Author Response

Response to Reviewer 1 Comments

Reviewer 1:

Point 1: This manuscript studied rice Oryza officinalis Wall tissue culture technology, observed pollen development, obtained callus and regenerated plantlets, and obtained some polyploid seedlings by colchicine treatment at 400 mg/L, the manuscript on flow cytometry data and field cultivation seedlings traits observation is very important, although the data is not quite perfect, I hope the author will make further observations, please transplant them into the text.

Response 1: Thanks for the kind suggestion. We have added the results of flow cytometry data and the observation of seedlings in the text to complete our results. 

Point 2: The manuscript writing is rough. Careful examination and revision are required.

Response 2: We agree with the reviewer and have revised and corrected the text throughout the manuscript. In addition, we let one native English speaker to revise the manuscript.     

Point 3: The rice Oryza officinalis Wall appears hundreds of times in the manuscript. It appears for the first time, the full name is required, after which it is referred to written as O. officinalis; The same is Oryza alta Swallen.

Response 3: Yes, we have revised the “Oryza officinalis Wall” and “Oryza alta Swallen” after it appeared in the first time.

Point 4: Spaces are needed in many places in the manuscript, such as between numbers and units, time and weight units, etc. Spaces are required before citations Instead of PGRs;

Response 4: Thanks. We added spaces in the manuscript and checked it carefully.

Point 5: The tissue culture process is not hormone, but plant growth regulators (PGRs); When plant growth regulators first appear, they need to be called in full, including in the abstracts and text, and then they can be abbreviated;

Response 5: Yes. We use plant growth regulators (PGRs) instead of the hormone. In addition, we listed the full name of plant growth regulators when it appeared at first instance.

Point 6: 6-BA changed as BA, KT changed as KIN; mg/L changed as mg.L-1, activated carbon changed as activated charcoal;

Response 6: Thanks for the kind suggestion. In the manuscript, 6-BA changed to BA, KT changed to KIN; mg/L changed to mg.L-1, and activated carbon changed to activated charcoal.

Point 7: To avoid using best in the manuscript, there is no best, only the change sign, it is recommended to change to “optimal”;

Response 7: Thanks for the kind suggestion. We have changed the “best” word in the manuscript to “optimal”.

Point 8: The reviewers marked the PDF version with a yellow marker, wish the authors can revise them.

Response 8: Thanks for the kind suggestion. We have corrected the text according to the PDF version with the yellow marker.

Point 9: Comments on the Quality of English Language. Need English improving

Response 9: Thanks for the kind suggestion. We have revised and corrected the text throughout the manuscript. In addition, we let one native English speaker revise the article. 

Reviewer 2 Report

Overall view: The article has not necessary standards to be published. It explains some incomplete experiments and results. The sources of interspecific hybrid and justifications should be explained well in the revised version.   

 Title

Please shorten Establishment of its Polyploidy Induction Tissue Culture System

Abstract

The abstract requires rewriting. At first, which method was used for interspecific hybrid production? The reason for each experiment, the explanation, and results of the research are not clear.

Introduction

Requires rearrangement and grammatical improvement. Also, some words need changing for example: cultivar rice to cultivated rice, …

 It is also needed to enrich is with adding some relevant literature. For example, the authors can address the following papers.

https://doi.org/10.1007/978-1-0716-1331-3_12

https://doi.org/10.21273/jashs.136.3.198

Results

The results are dispersed and explanations of figures do not differentiate the differences between cultivated rice, O. offficinalis and interspecific hybrid. Some words used in the images are technically correct (for example in Figure 1, “abnormal Telophase I” should be explained with the chromosome's karyotype pic; for microspore developmental stages the technical words are different).

Material and method

Incomplete explanation about interspecific hybrid: “The interspecific hybrid of wild rice (Oryza Officinalis Wall.) and cultivar rice was used in this study”?

The reference (if you have received form others) or method (if you have created this interspecific hybrid) and cytological verification (maybe it is an aneuploidy) is not explained. 

need to be edited by a native English speaker

Author Response

Response to Reviewer 2 Comments

Reviewer 2:

Point 1: Overall view: The article has not necessary standards to be published. It explains some incomplete experiments and results.

Response 1: Thanks. We have rewritten the abstract and cytological results in the text. In the revision, we added the flow cytometry data and the results of field cultivation seedlings into the text.

Point 2The sources of interspecific hybrid and justifications should be explained well in the revised version.   

Response 2: Thanks for the kind suggestion. We added the explanation of interspecific hybrid in the results, material and method section. The revision is listed as follows:

“This study used interspecific hybrid plants derived from O. officinalis and cultivated rice to investigate the reproductive character of interspecific rice hybrid plants constructed by O. officinalis (female parent, CC genome) and cultivated rice (male parent, AA genome)”.

Point 3Title: Please shorten Establishment of its Polyploidy Induction Tissue Culture System

 Response 3: Thanks for the kind suggestion. We have shortened the title and revised it as: “In Vitro Induction of Interspecific Hybrid Derived from Oryza officinalis Wall and Polyploidy Induction by Tissue Culture”.

Point 4Abstract: The abstract requires rewriting. At first, which method was used for interspecific hybrid production? The reason for each experiment, the explanation, and results of the research are not clear.

Response 4:  We agree with the reviewer and have rewritten the abstract according to the suggestion. The abstract's revision listed as follows:

Oryza officinalis Wall is a potential genetic resource for rice breeding; however, its distant genome limits its crossing ability with cultivated rice.  The interspecific hybridization of O. officinalis and cultivated rice, the establishment of its tissue culture, and the induction of polyploidy are ways to improve O. officinalis's poor crossability. In this study, we generated the interspecific hybrid and observed its reproductive character, and the results indicated that abortive pollens (81.94%) and embryo sac abnormalities (91.04%) were the primary cause of its high sterility. For callus induction of interspecific hybrid explants, two culture mediums [Chu's N-6 Medium (N6) and 1/2 Murashig and Skoog Medium (1/2 MS)], and four plant growth regulators [2,4-Dichlorophenoxyacetic acid (2,4-D), 6-Benzylaminopurine (BA), L-Proline (Pro) and acid hydrolyzed casein] were used. The optimal combination of N6+2,4-D (2.0 mg·L-1) produced the highest induction rate of 80.56%±5.44. For callus differentiation and proliferation, MS + BA (2.0 mg·L-1) + NAA (0.2 mg·L-1) produced the highest differentiation rate (75.00%±4.97) and seeding emergence ratio (28.97%±4.67). For root seedlings, 1/2 MS + NAA (2.0 mg·L-1) + BA (0.2 mg·L-1) was the optimal combination and produced an average number of 13.95  roots per plant. For polyploidy induction of interspecific hybrid, the concentration of colchicine treatment at 400 mg·L-1 for three days is the optimal protocol for inducing the polyploidy material. We established the tissue culture and polyploidy induction of interspecific hybrid, which was used to overcome the low cross ability of O. officinalis and transfer its desirable traits into cultivated rice.

Point 5Introduction: Requires rearrangement and grammatical improvement. Also, some words need changing for example: cultivar rice to cultivated rice.

Response 5: Thanks for indicating errors in the manuscript. We rearranged the introduction's content and improved the introduction section's grammar.  

Point 6It is also needed to enrich is with adding some relevant literature. For example, the authors can address the following papers. https://doi.org/10.1007/978-1-0716-1331-3_12 and https://doi.org/10.21273/jashs.136.3.198

 Response 6: Thanks for the kind suggestion. We have added the relevant literature in the revision.

Point 7Results: The results are dispersed and explanations of figures do not differentiate the differences between cultivated rice, O. offficinalis and interspecific hybrid.

Response 7: Thanks for the kind suggestion. We have rewritten the cytological results of the interspecific hybrid and its parent in the text. In addition, we also revised the explanations of the figures.

Point 8Some words used in the images are technically correct (for example in Figure 1, “abnormal Telophase I” should be explained with the chromosome's karyotype pic; for microspore developmental stages the technical words are different.

Response 8: Thanks for indicating errors in the article. We have revised cytological results and legend of figures. For example, the figure of the pollen development process of interspecific hybrid is revised as follows: “Pollen development process of interspecific hybrid. (a) Pre-Meiotic stage, the arrow indicates degraded pollen mother cell. (b) Telophase I, the arrow indicates asynchronous pollen). (c) Telophase II, the arrow indicates abnormal tetrad cell, triad. (d) Tetrad stage, arrows indicate abnormal tetrad cells, two triad. (e) Tetrad stage, the arrow indicates the abnormal shape of tetrad). (f) Single microspore stage, degradation of microspore cells. (g) Middle single microspore stage, arrows indicate the abnormal shape of microspores. (h) Mature pollen stage, the arrow indicates spherical abortive pollen. (i) Mature pollen stage, arrows indicate typical abortive pollens). Bar=40 μm”.

Point 9Material and method. Incomplete explanation about interspecific hybrid: “The interspecific hybrid of wild rice (Oryza Officinalis Wall.) and cultivar rice was used in this study”?

Response 9: Thanks for the useful suggestion. We have added the explanation of interspecific hybrid.

Point 10The reference (if you have received form others) or method (if you have created this interspecific hybrid) and cytological verification (maybe it is an aneuploidy) is not explained. 

 Response 10:  Yes, we have explain the interspecific hybrid in the cytological results section and method section. In addition, we also added the related references in the introduction and discussion section.  

Round 2

Reviewer 1 Report

There are two inductions in the headline, and it is suggested that the title be changed to: In Vitro Induction of Interspecific Hybrid and Polyploidy Derived from Oryza officinalis Wall;

L26, The optimal N6 medium supplemented with 2.0 mg· L-1 2,4-D;

L 27, MS medium supplemented with 2.0 mg· L-1 BA + 0.2 mg· L-1 NAA;

samely, L29, L161, L330;

L27, percentage changed to (80.56 ± 5.44)%, same L160, et al.;

Consecutive citation numbers changed to L42, [2-4], L72, [19-23], L340 [39-41];

Table 2, Differential analysis results appeared to be problematic

L385, how long with 75% alcohol, 0.1% HgCl2 disinfection, Clean with? (Distill water?)

It is OK.

Author Response

Response to Reviewer 1 Comments

Reviewer 1:

Point 1: There are two inductions in the headline, and it is suggested that the title be changed to: In Vitro Induction of Interspecific Hybrid and Polyploidy Derived from Oryza officinalis Wall;

Response 1: Thanks for the kind suggestion. We have changed the title as “In Vitro Induction of Interspecific Hybrid and Polyploidy Derived from Oryza officinalis Wall”;

Point 2: L26, The optimal N6 medium supplemented with 2.0 mg· L-1 2,4-D;

Response 2: Yes. We have changed “The optimal combination of N6 + 2,4-D (2.0 mg·L-1) ” in the manuscript to “The optimal N6 medium supplemented with 2.0 mg· L-1 2,4-D“.   

Point 3: L 27, MS medium supplemented with 2.0 mg· L-1 BA + 0.2 mg· L-1 NAA;

Response 3: Yes. We have changed “MS + BA (2.0 mg·L-1) + NAA (0.2 mg·L-1) ” in the manuscript to “MS medium supplemented with 2.0 mg· L-1 BA + 0.2 mg· L-1 NAA“.   

Point 4: L27, percentage changed to (80.56 ± 5.44)%, same L160, et al.;

Response 4: Thanks. We have changed “80.56% ± 5.44” to (80.56 ± 5.44)% in line 27, including the L160 and et al.

Point 5: Consecutive citation numbers changed to L42, [2-4], L72, [19-23], L340 [39-41];

Response 5: Thanks for the kind suggestion. We have changed L42, L72, and L340 to [2-4], [19-23], and [39-41], respectively.  

Point 6: Table 2, Differential analysis results appeared to be problematic?

Response 6: Sorry, it is a mistake. We have revised the differentiation rate in F2 combination, the really rate is “50.56% ± 6.25”.  

Point 7: L385, how long with 75% alcohol, 0.1% HgCl2 disinfection, Clean with? (Distill water?)

Response 7: Yes, We have revised the method according to suggestion, the revision was listed as follow:

“The panicles were surface-sterilized the 75% ethanol for 30 s, and then disinfected in 0.1% HgCl2 (mercury chloride) solution 8-10 min. Then, samples were cleaned with the sterile distilled water for 4-6 times”.

Reviewer 2 Report

The authors revised the manuscript according to the reviewers' comments thoroughly and can be accepted

Fine

Author Response

Response to Reviewer 2 Comments

Reviewer 2:

Point 1: The authors revised the manuscript according to the reviewers' comments thoroughly and can be accepted

Response 1: We are very thankful for the reviewer for encouraging comments about the manuscript to publish in Plants.